# Delta Radiomics and Tumor Size: A New Predictive Radiomics Model for Chemotherapy Response in Liver Metastases from Breast and Colorectal Cancer

**DOI:** 10.3390/tomography11030020

**Published:** 2025-02-20

**Authors:** Nicolò Gennaro, Moataz Soliman, Amir A. Borhani, Linda Kelahan, Hatice Savas, Ryan Avery, Kamal Subedi, Tugce A. Trabzonlu, Chase Krumpelman, Vahid Yaghmai, Young Chae, Jochen Lorch, Devalingam Mahalingam, Mary Mulcahy, Al Benson, Ulas Bagci, Yuri S. Velichko

**Affiliations:** 1Department of Radiology, Feinberg School of Medicine, Northwestern University, Chicago, IL 60611, USA; nicolo.gennaro@northwestern.edu (N.G.); moataz_as20@hotmail.com (M.S.); amir.borhani@nm.org (A.A.B.); linda.kelahan@nm.org (L.K.); hatice.savas@nm.org (H.S.); ryan.avery@nm.org (R.A.); kamal.subedi@nm.org (K.S.); tugce.agirlartrabzonlu@nm.org (T.A.T.); chase.krumpelman@nm.org (C.K.); ulas.bagci@northwestern.edu (U.B.); 2Department of Radiological Sciences, University of California Irvine, Irvine, CA 92868, USA; vyaghmai@hs.uci.edu; 3Department of Medicine, Division of Hematology/Oncology, Feinberg School of Medicine, Northwestern University, Chicago, IL 60611, USA; ychae@nm.org (Y.C.); jochen.lorch@nm.org (J.L.); mahalingam@nm.org (D.M.); mary.mulcahy@nm.org (M.M.); albenson@nm.org (A.B.); 4Robert H. Lurie Comprehensive Cancer Center, Northwestern University, Chicago, IL 60611, USA

**Keywords:** computer tomography, liver, metastases, radiomics, Delta radiomics, chemotherapy, response assessment, RECIST 1.1

## Abstract

**Background/Objectives**: Radiomic features exhibit a correlation with tumor size on pretreatment images. However, on post-treatment images, this association is influenced by treatment efficacy and varies between responders and non-responders. This study introduces a novel model, called baseline-referenced Delta radiomics, which integrates the association between radiomic features and tumor size into Delta radiomics to predict chemotherapy response in liver metastases from breast cancer (BC) and colorectal cancer (CRC). **Materials and Methods**: A retrospective study analyzed contrast-enhanced computed tomography (CT) scans of 83 BC patients and 84 CRC patients. Among these, 57 BC patients with 106 liver lesions and 37 CRC patients with 109 lesions underwent post-treatment imaging after systemic chemotherapy. Radiomic features were extracted from up to three lesions per patient following manual segmentation. Tumor response was assessed by measuring the longest diameter and classified according to RECIST 1.1 criteria as progressive disease (PD), partial response (PR), or stable disease (SD). Classification models were developed to predict chemotherapy response using pretreatment data only, Delta radiomics, and baseline-referenced Delta radiomics. Model performance was evaluated using confusion matrix metrics. **Results**: Baseline-referenced Delta radiomics performed comparably or better than established radiomics models in predicting tumor response in chemotherapy-treated patients with liver metastases. The sensitivity, specificity, and balanced accuracy in predicting response ranged from 0.66 to 0.97, 0.81 to 0.97, and 80% to 90%, respectively. **Conclusions**: By integrating the relationship between radiomic features and tumor size into Delta radiomics, baseline-referenced Delta radiomics offers a promising approach for predicting chemotherapy response in liver metastases from breast and colorectal cancer.

## 1. Introduction

The liver is the most common site of metastatic disease, with approximately 30% of colorectal cancer (CRC) patients and 50% of breast cancer (BC) patients presenting with metastatic disease to the liver at the time of diagnosis. Up to 80% of patients with liver metastatic disease are not candidates for liver resection or local ablative treatments; hence, the standard first-line treatment in majority of cases is systemic chemotherapy, with the intention of reducing metastatic burden in preparation for surgical resection or to prolong survival in the event of extended disease.

Tumor heterogeneity is a leading cause of cancer recurrency, therapeutic resistance, and poor overall survival [1,2,3,4]. Biopsy is not feasible for the comprehensive evaluation of tumor heterogeneity in most cases, as it requires the sampling of all individual lesions and is subject to sampling error [5]. Imaging provides a non-invasive and more comprehensive alternative for assessing the local tumor environment. This can be performed using computer tomography (CT), which quantifies variations in atomic density; magnetic resonance imaging (MRI), which evaluates the physical properties of water and fat protons in the cancer tissue; or positron emission tomography (PET), which allows for metabolic analysis of tumor sub-habitats [6,7]. Advances in imaging and post-processing techniques have led to increasingly high-throughput image analysis. Radiomics has emerged as a new framework that expands on conventional radiological semantic assessment by extracting more comprehensive tissue characteristics based on quantifying tumor image heterogeneity and microenvironment [8,9,10,11].

Liver metastases from breast and colorectal cancer present a significant clinical challenge, and early assessment of response to systemic therapy is crucial for optimal patient management. Delta radiomics, which analyzes changes in radiomic features between pre- and post-treatment images, offers a promising approach for monitoring and predicting therapy response. By quantifying the changes in tumor characteristics captured through radiomic feature analysis of paired pre- and post-treatment medical images [12,13,14,15], this technique has the potential to identify imaging biomarkers that enable earlier and more accurate prediction of treatment outcomes. This, in turn, could lead to more personalized and effective treatment strategies for patients with metastatic breast and colorectal cancer. Several recent studies have explored the application of machine learning and CT Delta radiomics to predict tumor response in colorectal and breast cancer liver metastases treated with chemotherapy or targeted therapy (see Appendix A for a summary of key studies) [16,17]. For example, Ye et al. [18] demonstrated the potential of Delta radiomics, analyzing changes in CT features before and after chemotherapy, to improve the prediction of progression-free survival in patients with colorectal liver metastases, particularly when combined with clinical characteristics. Their model, based on Delta radiomics and clinical data, outperformed models based on single time-point radiomics. Similarly, Lu et al. [19] showed that a deep learning network can effectively capture tumor morphological changes beyond size, improving early prediction of treatment response to anti-VEGF therapy in metastatic colorectal cancer and outperforming traditional size-based assessments. Integrating this deep learning approach with conventional size measurements further enhanced predictive accuracy, suggesting its value for personalized early treatment decisions in metastatic colorectal cancer. These findings, while promising, highlight the need for further research to validate the potential of Delta radiomics for early response assessment and personalized treatment strategies, particularly in breast cancer liver metastases.

Despite the progress made in establishing radiomics in clinical studies, this area of research remains challenging and demanding. For example, only a few studies have explored the correlation between tumor heterogeneity and size [20,21,22,23,24], even though it is widely accepted that heterogeneity increases as tumors grow [25,26,27,28,29]. Tumor size is a critical feature of tumor phenotype for local staging, while tumor shrinkage is widely considered a surrogate endpoint of therapy efficacy. In fact, the Response Evaluation Criteria in Solid Tumors (RECIST) are the most used anatomic imaging biomarkers in clinical trials involving various cancers [30,31]. Therefore, developing a radiomics framework that considers changes in tumor texture associated with tumor size has the potential to improve the assessment of tumor heterogeneity, provide new insights into tumor biology, and enhance the precision of therapy response evaluation, potentially uncovering novel predictive biomarkers.

Our contribution can be summarized as the following:

The Introduction of a Novel Baseline-Referenced Delta Radiomics Model: The core novelty lies in proposing a new type of Delta radiomics model that eliminates the need for individual patient baseline radiomics data. This is achieved through a baseline-referencing approach.The Utilization of Large Pretreatment Datasets to Establish Baseline Relationships: The model leverages large datasets of pretreatment scans to empirically derive the functional relationships between radiomic features and tumor size. This data-driven approach allows for a robust and generalizable baseline reference.Integration of the Radiomic Feature–Tumor Size Relationship into Delta Radiomics: The study incorporates the functional association between radiomic features and tumor size directly into the Delta radiomics framework.Enabling Delta Radiomics with Post-Treatment Scans Alone: A significant practical contribution is the model’s ability to calculate Delta radiomics values and assess therapy response using only post-treatment scans. This overcomes a key limitation of traditional Delta radiomics, which requires both pre- and post-treatment imaging.Validation Against Conventional Radiomics Models: The study provides empirical validation by comparing the performance of the novel baseline-referenced Delta radiomics model against established methods: pretreatment radiomics and conventional Delta radiomics. This comparative analysis demonstrates the potential advantages of the proposed approach in a clinically relevant setting (liver metastases and systemic therapy response prediction).

This paper first establishes the Materials and Methods, then describes the response assessment models, including the novel baseline-referenced Delta radiomics model. It details the development of the predictive model and the evaluation methods used to assess its performance. Finally, the paper concludes by discussing the results, clinical implications, future directions, and limitations of this project.

## 2. Materials and Methods

This retrospective study was approved by the Institutional Review Board and was granted a waiver of Health Insurance Portability and Accountability Act authorization and of written informed consent. The research methods were performed in accordance with the Declaration of Helsinki. Figure 1 provides a summary of the study workflow, while Figure 2 illustrates the steps involved in building the datasets. A part of our dataset has been used for another study [24].

### 2.1. Patient Cohort

The study focused on patients diagnosed with BC and CRC at a single enterprise institution with multiple sites between 2016 and 2022.

The inclusion criteria for patient selection were as follows: (a) pathologic confirmation of breast or colorectal cancer as the primary disease, (b) pathologic confirmation of hepatic metastasis from breast or colorectal cancer, (c) preoperative contrast-enhanced abdominal CT performed within one month from the start of systemic therapy, and (d) availability of first CT follow-up after medical treatment. The exclusion criteria were as follows: (a) previous history of cancer or other local or systemic treatments, (b) interval longer than 5 months between baseline study and follow-up, (c) maximum diameter <10 mm or infiltrative, non-segmentable lesions, (d) diffuse background liver disease (i.e., cirrhosis, depositional liver disease), and (e) absence of portal venous phase or poor-quality radiologic studies.

The search resulted in two datasets. The first dataset consisted of 83 BC and 84 CRC patients with 155 and 192 liver metastases, respectively, with pretreatment CT scans. Of these patients, 57 BC and 37 CRC patients with 106 and 109 liver lesions underwent post-treatment CT imaging, forming the second dataset. The first dataset was also used for radiomics data harmonization, with additional pretreatment CT imaging of 228 patients with 280 hepatic cysts and 707 samples of normal liver parenchyma [24]. Clinical data for both datasets, such as age, sex, lesion size, and follow-up period, can be found in Table 1. Information regarding the chemotherapy regimens used in the study is available in the Appendix A.

Patients underwent contrast-enhanced abdominal CT on six different Siemens (Erlangen, Germany) scanners: Sensation 64, Somatom Definition, Somatom Definition AS, Somatom X.cite, Somatom Force, Somatom Drive. These scanners utilized variable acquisition techniques and protocols during the imaging process. Only portal venous phase scans were considered for segmentation and radiomics analysis. Technical details of CT scanners are available in our previous publication [24]. Scanner information is available in the (Appendix A).

Our radiomics framework involves five main steps: (1) CT preprocessing; (2) tumor segmentation; (3) CT radiomics radiomic feature extraction; (4) radiomic feature harmonization; and (5) classification, where we trained machine learning models. The following sections provide a detailed description of each stage.

### 2.2. CT Preprocessing

Volumetric radiomics analyses following IBSI recommendations were conducted. Three-dimensional spatial resampling of CT scans, using a fifth-degree Lagrangian polynomial interpolation, was performed to achieve an isovoxel size of 1 × 1 × 5 mm^3^ within the ROI [24].

### 2.3. Tumor Segmentation and Response Assessment

Volumetric lesion segmentation was performed manually using LIFEx software, version 6.30 (CEA, Orsay, France) [32] on axial plane (3–5 mm slice thickness) by one of two radiologists (N.G. and M.S.), with 6 and 3 years of experience in abdominal radiology and 2 years of experience in quantitative and radiomics analysis, respectively. Following segmentation, the tumor delineations for each lesion were reviewed. Discrepancies were resolved by N.G. Up to three well-defined and well-separated liver metastases with the longest diameter larger than 10 mm were selected in each patient. A range of 2–3 mm of perilesional rim was included in the volume of interest.

RECIST 1.1 provides an overall response assessment of disease by measuring changes in tumor burden, which is the sum of selected target lesion sizes. However, this approach is not suitable for radiomics models since tumor texture varies depending on the tissue type, and radiomics features computed from different organs cannot be combined as additive values. To address this limitation, we applied RECIST 1.1 for a per-organ (liver) treatment response assessment. For the assessment of the objective response, we measured the longest diameter of each lesion in the axial plane. We categorized the objective response into three groups: progressive disease (PD), stable disease (SD), and partial response (PR) [30], with at least a 20% increase in tumor burden for progressive disease and at least a 30% shrinkage for partial response. Lesions that showed changes within the range of +20% to −30% were categorized as SD.

### 2.4. CT Radiomics Feature Extraction

The absolute gray-level discretization was performed within the volume of interest in the range of intensity values varying from −50 Hounsfield Unit (HU) to 300 HU using a fixed number of bins equal to 36 and a fixed bin size equal to 10 HU. Three-dimensional CT-based radiomics features from the intensity-based statistics (7 features), intensity histogram (3 features), discretized intensity statistics (10 features), discretized intensity histogram (6 features), shape features (4 features including volume), gray-level co-occurrence matrix (GLCM, 7 features), gray-level run length matrix (GLRLM, 11 features), neighborhood gray-level different matrix (NGLDM, 3 features), and gray-level zone size matrix (GLZSM, 11 features) classes were extracted using LIFEx software, version 6.30 (Orsay, France) [32,33]. Appendix A provides a list of all the radiomic features used in this study.

### 2.5. CT Radiomic Feature Harmonization

To standardize the radiomics features across different CT scanners (Table 1), a linear mixed-effects model (LME) for radiomics data harmonization was applied [24]. LME analysis was performed to assess the impact of three fixed-effect variables: cancer/tissue type (*T*), scanner (*S*), and tumor volume (*V*). The volume was the primary variable of interest. Cancer/tissue type and scanner were treated as fixed factors, influencing both the intercept and slope of the linear model. The LME model can be summarized as follows:(1)$$F_{T,S}=f^{R}+r_{T}+r_{S}+\left(\Delta f^{R}+\Delta r_{T}+\Delta r_{S}\right)\cdot V$$

Equation (1) models the linear relationship between a radiomics feature and tumor volume (Figure 3). This model includes fixed effects, which define the reference group, $F^{R}=f^{R}+\Delta f^{R}\cdot V$, where $f^{R}$ is the intercept and $\Delta f^{R}$ is the slope. The model also incorporates random effects for scanner and tissue type, including random intercepts, $r_{S}$ and $r_{T}$, and random slopes, $\Delta r_{S}$ and $\Delta r_{T}$.

After estimating all random effects, the radiomics features were harmonized by removing the scanner-related random effects:(2)$$F_{T}^{H}=F_{T,S}-r_{S}-\Delta r_{S}\cdot V$$
where $F_{T}^{H}$ represents the harmonized radiomics feature, adjusted to remove scanner effects and referenced to the designated group. This harmonization process removes scanner-related variability, preserving the association between radiomics features and tumor size specific to each cancer type. It is important to note that while this approach facilitates more reliable comparisons and analyses of radiomics data, it provides limited control over the broader range of imaging and other technical parameters.

### 2.6. Model Building and Statistical Analysis

To investigate the relationship between radiomics features and tumor volume (Figure 3), we first analyzed a large dataset of pretreatment scans [24]. We employed five fitting models (linear, logarithmic, inverse, power, and exponential) to assess these relationships [21,22,23]. The goodness-of-fit for each model was evaluated using the coefficient of determination (R-squared). The best-fitting model for each feature was selected based on the highest R-squared value and the condition that the model residuals passed a normality test [23]. The function derived from the best-fitting model was then used as a baseline reference for further analysis.

To evaluate the response to treatment, we constructed three radiomics models. The first model, the pretreatment model, employed only the pretreatment (baseline) radiomics features. For this model, all available pretreatment radiomic features were considered. The second model, Delta radiomics, utilized both the pre- and post-treatment radiomics features [34] and was trained on Delta radiomics features calculated using the following equation:(3)$$\Delta RF=\frac{RF_{post}-RF_{pre}}{RF_{pre}},$$
where $RF_{pre}$ and $RF_{post}$ are pre- and post-treatment radiomics features, respectively. All available radiomic features extracted from both pre- and post-treatment scans were used to calculate the Delta radiomics features. The third model, baseline-referenced Delta radiomics, was built exclusively on post-treatment scans, substituting the pretreatment radiomics data for each individual patient with a corresponding baseline reference function. This baseline reference function was derived for each radiomic feature individually. Specifically, for each patient, the pretreatment value of a given radiomic feature served as the baseline reference for that feature. The baseline-referenced Delta radiomic features were calculated similarly to Delta radiomics features:(4)$$\Delta RF_{BR}=\frac{RF_{post}-RF_{BR}\left(V\right)}{RF_{BR}\left(V\right)},$$
where $RF_{BR}\left(V\right)$ is the baseline reference function for a given radiomics feature V, representing the pretreatment value of that feature for the specific patient. All available radiomic features extracted from the post-treatment scans, in conjunction with their corresponding baseline references, were used to generate the baseline-referenced Delta radiomics features. The rationale behind this approach is to normalize the post-treatment radiomic features with respect to each patient’s baseline status for that feature, allowing for a more personalized assessment of change.

The pretreatment model, using only baseline radiomics features, serves as a valuable reference point for evaluating the performance of more complex response assessment models like the Delta radiomics and baseline-referenced Delta radiomics models. While the pretreatment model provides a baseline understanding of the predictive power of pretreatment features alone, it is important to note that it does not predict treatment response on its own. Instead, it helps to establish the incremental value of incorporating post-treatment features, as carried out in the Delta radiomics and baseline-referenced Delta radiomics models, for improved response assessment.

The Random Forest, LogitBoost, svmRadial, and pcaNNet models were used to develop classification models for predicting response to chemotherapy and assessing feature importance. Each model incorporates all available transformed features (i.e., the pretreatment features for the pretreatment model, the Delta radiomics features for the Delta radiomics model, and the baseline-referenced Delta radiomics features for the baseline-referenced Delta radiomics model). These models employ diverse strategies for feature computation and combination. These models employ diverse strategies for feature computation and combination. Random Forest constructs an ensemble of decision trees, each trained on a random subset of features and data and aggregates their predictions (effectively weighting feature importance based on their contribution to the ensemble). LogitBoost, a boosting algorithm, iteratively builds an ensemble of weak learners (typically decision stumps), weighting them based on their performance and focusing on misclassified instances. svmRadial maps all features to a high-dimensional space using a radial basis function kernel and identifies an optimal hyperplane to separate classes. pcaNNet uses principal component analysis to reduce the dimensionality of the feature space (while still incorporating information from all features) before training a neural network, thereby addressing potential issues with high dimensionality and multicollinearity. Feature importance was assessed using the Loess R-squared coefficient; a higher value indicates greater feature importance in the classification model.

The whole dataset was partitioned into a training set (75%) and a test set (25%). A comparison of the predicted and actual responses was determined by creating a confusion matrix. The performance of the model was determined by calculating the sensitivity, specificity, positive predictive value (PPV), negative predicative value (NPV), and balanced accuracy. Receiver operating characteristic (ROC) and area under the curve (AUC) ROC analyses were also conducted to evaluate the model’s discriminative ability. Bootstrapping with 1000 iterations was performed to estimate the model’s performance and 95% confidence intervals. Analysis was performed using the Caret package (Pfizer Global R&D, Groton, CT, USA) [35] using the statistical environment R version 4.2.2 (Vienna, Austria) [36]. A p-value of less than 0.05 was considered statistically significant.

## 3. Results

### 3.1. Patient Characteristics

After applying the inclusion and exclusion criteria, the final cohort comprised of 57 patients with BC with a mean age of 57 ± 13 years and 37 patients with CRC (19 females and 18 males) with a mean age of 60 ± 13 for response assessment analysis. The BC group included 106 liver metastasis lesions, and the CRC group included 109 lesions.

### 3.2. Objective Response Assessment

The mean tumor diameter was 2.3 ± 1.3 cm for BC and 3.1 ± 1.5 cm for CRC patients, respectively (Table 1). The median interval and the interquartile range (IRQ) between imaging assessments was 108.5 (78–131) days for BC patients and 75.0 (58–103) days for CRC patients, respectively. According to RECIST 1.1 criteria, 22 (20%) lesions in CRC patients were classified as PD, 70 (64%) as SD, and 18 (16%) as PR. In BC patients, 30 (28%) lesions were classified as PD, 57 (54%) as SD, and 19 (18%) as PR.

### 3.3. Models’ Performance for Response Assessment

Figure 4 shows the confusion matrices for each model calculated using the Random Forest model. Pretreatment radiomics exhibited weaker correlations with patient outcomes compared to the Delta radiomic and baseline-referenced Delta radiomic models. In contrast, the Delta radiomic and baseline-referenced Delta radiomic models demonstrated better predictive performance across all classes (PR, SD, and PD) for both CRC and BC patients. Notably, the baseline-referenced Delta radiomic model consistently showed higher correlations than the Delta radiomic model across both cancer types. These metrics are summarized in Table 2. The performance of other machine learning models (LogitBoost, svmRadial, and pcaNNet) for response assessment in patients with CRC and BC liver metastases is summarized in Appendix A.

Figure 5 displays the ROC curves for all models. In CRC, both the Delta radiomic and baseline-referenced Delta radiomic models significantly outperformed the pretreatment radiomics model. The baseline-referenced Delta radiomic model offered a slight advantage over the Delta radiomic model, particularly at lower false positive rates. The area under the ROC curve (AUC) values for CRC were 0.82 ± 0.08 and 0.93 ± 0.04 for the Delta radiomic and baseline-referenced Delta radiomic models, respectively. Similarly, in BC, both Delta radiomic models showed substantial improvement over the pretreatment radiomics model. However, the difference between the Delta radiomic and baseline-referenced Delta radiomic models was minimal. The AUC values were 0.90 ± 0.04 and 0.91 ± 0.04 for the Delta radiomic and baseline-referenced Delta radiomic models, respectively.

### 3.4. Variable Importance

Figure 6 shows the significance of each radiomic feature for the pretreatment, Delta radiomics, and baseline-referenced Delta radiomics models after bootstrap averaging. Notably, significant features identified by the Delta and baseline-referenced Delta radiomics models exhibited strong correlations in both the CRC and BC groups, whereas correlations with the pretreatment model were minimal. Specifically, GLRLM_RLNU, NGLDM_Coarseness, and GLZLM_GLNU occurred as the top three significant features for both the Delta and baseline-referenced Delta radiomics models in the CRC and BC groups. This suggests that these features are robust indicators of treatment response, consistent across both model types. Furthermore, Figure 7 demonstrates a strong association between changes in each of these three features and response categories. This visual representation supports the clinical relevance of these radiomic features in predicting treatment outcomes.

We also investigated differences between treatment groups within the CRC cohort, which included patients receiving either chemotherapy alone or chemotherapy in combination with bevacizumab. Bevacizumab, an anti-VEGF agent, targets tumor vascularization and is therefore expected to influence contrast enhancement patterns on CT. Appendix A provides detailed information about the chemotherapy treatments received by all patients. Figure 8 illustrates changes in several radiomic features—computed using the Delta radiomics and baseline-referenced Delta radiomics models—across treatment groups and response categories. Specifically, the median lesion intensity (CONVENTIONAL_HUQ2) showed a notable difference between treatment groups within the PD category. This suggests that bevacizumab may be associated with reduced contrast enhancement in progressive disease, potentially reflecting its anti-angiogenic effects. Other radiomic features, such as GLRLM_LGRE and GLZLM_LGZE, also show a similar tendency within the PD group. These features indicate changes in tumor heterogeneity. This suggests that bevacizumab not only reduces contrast enhancement but also affects the structural complexity of tumors in progressive disease, highlighting its potential impact on tumor biology.

## 4. Discussion

Tumor size is a cornerstone of tumor response assessment, but it may not be sufficient to indicate early disease progression or treatment response. Radiomics, a new source of quantitative imaging biomarkers, can predict tumor response and complement conventional response assessment metrics. Unlike size-based metrics, longitudinal radiomics data provide more comprehensive information about changes in each lesion over time. This can be valuable for understanding disease progression, evaluating treatment response, and predicting outcomes. However, obtaining longitudinal imaging data for cancer patients can be challenging due to various factors, including patient compliance, the availability and consistency of imaging, and the need for frequent imaging sessions that may not always be practical or feasible. These limitations can result in incomplete or inconsistent data, making it difficult to fully understand the dynamics of disease progression and treatment response over time. Despite these challenges, large imaging datasets contain a wealth of information that can be used to gain insight into various aspects of the disease and improve clinical decision making.

This study investigates the relationship between pretreatment radiomic features and tumor size in the context of treatment response assessment. As tumors grow, they become more heterogeneous due to increased cellular proliferation, necrosis, and alterations in the tumor microenvironment. These changes are reflected in both the tumor’s radiographic appearance and its radiomic features. However, the relationship between tumor texture and size can be influenced by factors such as tumor type, location, and prior treatment. To address these factors and establish baseline references, we analyzed pretreatment CT scans of liver metastases from both BC and CRC using large datasets. We then introduced a novel radiomics framework, baseline-referenced Delta radiomics, which leverages post-treatment radiomics data to assess therapy response by using baseline-referenced functions representative of the pretreatment state instead of actual pretreatment radiomics data. Notably, this approach allows for response assessment even in the absence of baseline imaging. While these baseline-referenced functions provide a valuable reference related to the patient’s pretreatment condition, it is important to acknowledge that they may not fully capture the complexity of the disease or serve as a complete substitute for longitudinal data. Despite these limitations, this model offers two potential clinical advantages. First, it could enable earlier assessment of treatment response compared to traditional methods like RECIST, potentially leading to more timely treatment adjustments. Second, it could be particularly valuable in resource-limited settings where consistent baseline imaging is not always feasible. For example, in cases where the model predicts a poor response early on, clinicians might consider switching to an alternative therapy or exploring clinical trials. In resource-constrained environments, the ability to assess response without baseline scans could expand access to personalized medicine. It is important to note, however, that the model’s performance and clinical utility need to be validated in larger, prospective studies before it can be widely adopted in clinical practice.

To evaluate the effectiveness of our proposed model, we compared the performance of pretreatment radiomics, Delta radiomics, and baseline-referenced Delta radiomics in assessing chemotherapy response in patients with colorectal and breast cancer liver metastases. The Delta radiomics model, which considers longitudinal changes in tumor texture, demonstrated high accuracy in response assessment for both cohorts. As expected and consistent with previous studies, the pretreatment radiomics model performed poorly [18]. The baseline-referenced Delta radiomics model achieved similar performance to the Delta radiomics model in both cohorts. Analysis of both Delta radiomics models revealed high correlations in significant features, demonstrating the effectiveness of using baseline-referenced functions to predict tumor response. The lower accuracy observed for both Delta radiomics models in the BC cohort may be attributable to the greater heterogeneity of treatments received (Appendix A) and the challenges associated with segmenting hepatic metastases in BC (e.g., less distinct margins and irregular shapes). Furthermore, the varied treatment regimens, particularly within the breast cancer cohort, may influence our conclusions regarding chemotherapy response. For example, if one regimen dominates in the dataset, the model’s performance might reflect its specific efficacy rather than a generalizable response to chemotherapy. On the other hand, a highly diverse range of treatments could obscure true associations between radiomic changes and response due to the confounding effects of the treatments themselves.

To assess the robustness of these findings across different machine learning algorithms, we performed an additional analysis using LogitBoost, svmRadial, and pcaNNet models. All models demonstrated similar trends in performance between pretreatment radiomics, Delta radiomics, and baseline-referenced Delta radiomics, mirroring the results observed with Random Forest. Random Forest and LogitBoost demonstrated similar performances, with Random Forest exhibiting slightly higher balanced accuracy. The svmRadial and pcaNNet models achieved balanced accuracies 5–15% lower than Random Forest. This difference may be attributable to the specific characteristics of these algorithms, such as the kernel function used in svmRadial or the potential impact of the principal component step in pcaNNet on the data representation. These results suggest that our primary findings regarding the effectiveness of Delta radiomics are consistent across a range of machine learning approaches.

Recent studies have explored the application of machine learning and CT Delta radiomics to predict tumor response in CRC and BC liver metastases treated with chemotherapy or targeted therapy (Appendix A summarizes several key studies) [16,17]. While many studies have investigated factors affecting texture features, few have examined the relationship between tumor size and heterogeneity [21,22,23] and even fewer have considered this relationship significant for response assessment. We demonstrate the effectiveness of both Delta radiomics models in evaluating treatment effects, for example, the difference between chemotherapy alone and chemotherapy in combination with bevacizumab in CRC patients. To our knowledge, this study is the first to derive a Delta radiomics score using only post-treatment imaging.

Our study had several limitations. First, it was a retrospective study with a limited sample size and heterogeneous treatment regimens, especially within the BC cohort. This retrospective design, while reflecting real-world clinical practice, introduces the possibility of selection bias and limits our ability to fully control for confounding variables. Second, we only considered CT scans of liver metastases from BC and CRC. While CT is a readily available and commonly used imaging modality for assessing liver metastases, it has inherent limitations in fully characterizing all tumor features. Certain radiological characteristics may not be optimally represented on CT, potentially leading to an incomplete assessment of the tumor response to therapy. In future, the model analysis should be further expanded to include other imaging modalities, such as MRI and PET, which may provide complementary information. Furthermore, our study relied on RECIST 1.1 for response assessment, criteria which have known limitations and may not fully capture the complex changes occurring within liver metastases, potentially impacting the sensitivity of treatment response assessment in this specific setting. Although we applied this method on CT scans, the concept of baseline-referenced Delta radiomics can be translated to other imaging modalities, including MRI and PET. The imaging data were collected from different medical centers, and the acquisition protocols and scanners used were inevitably different among them, which may have affected the radiomics features. However, the rigorous harmonization process applied in our study allowed us to successfully address this acquisition heterogeneity [24]. While the sample size provides statistical power for detecting key trends, the variability in patient characteristics and treatment regimens could influence the observed associations between radiomic features and treatment response, potentially limiting the generalizability of the findings. Additionally, the relatively small sample size, coupled with potential external influences such as variations in treatment methods and patient demographics (including race, region, and physical fitness), limits the generalizability of our findings and warrants further investigation in larger, more diverse cohorts. In this study, we did not employ imaging filters such as Gaussian, Laws, or Wavelets. Future work could explore the impact of data augmentation on the relationship between radiomic features and tumor size, as well as investigate optimal combinations of augmentation techniques and feature selection methods. This would improve the robustness and generalizability of Delta radiomics models across diverse datasets and imaging protocols. At present, there are no published results on the analysis of longitudinal changes in radiomics features that can be used to compare their association with tumor size. Therefore, our conclusions are limited to a general discussion.

## 5. Conclusions

In conclusion, the concept of baseline-referenced Delta radiomics has shown promising results in predicting therapy response. This approach can potentially complement traditional size-based metrics for response assessment and enhance our understanding of disease progression and treatment response. These findings encourage the application of baseline-referenced Delta radiomics in several areas, including longitudinal studies to predict long-term outcomes, response durability, and the emergence of resistance mechanisms. Furthermore, baseline-referenced radiogenomics has the potential to identify associations between imaging features and underlying molecular characteristics linked to treatment response, helping to identify potential therapeutic targets. Further validation in larger, prospective cohorts is warranted to confirm these findings and explore the clinical utility of this approach for personalized treatment strategies.

## Figures and Tables

**Figure 1 tomography-11-00020-f001:**
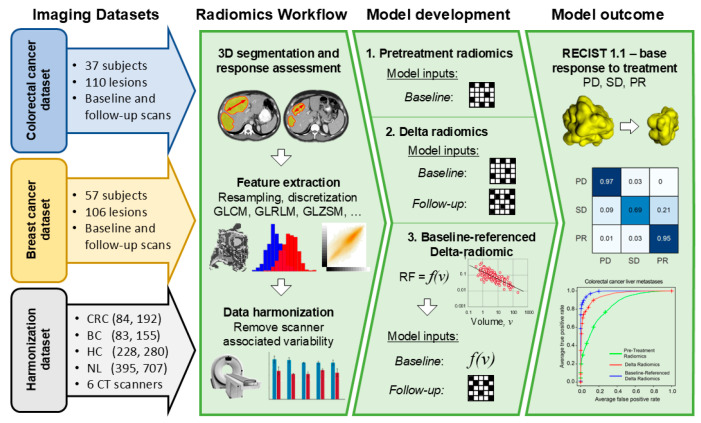
Pipeline of this study.

**Figure 2 tomography-11-00020-f002:**
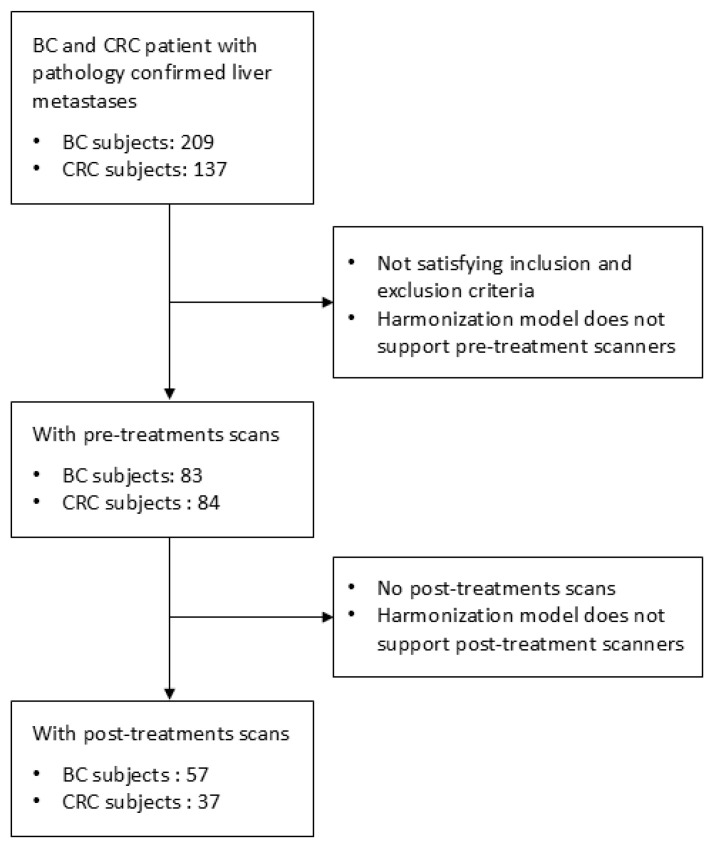
Flowcharts illustrate the construction of the datasets.

**Figure 3 tomography-11-00020-f003:**
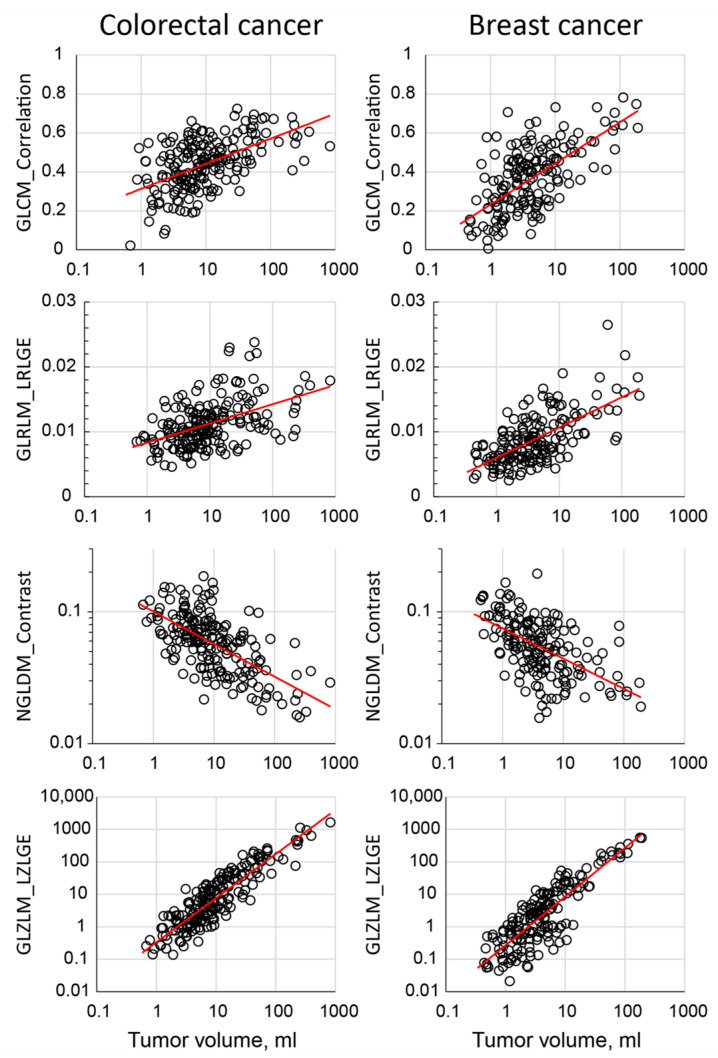
Scatter plot demonstration of association between radiomic features and the tumor size. The red line represents the fitted result.

**Figure 4 tomography-11-00020-f004:**
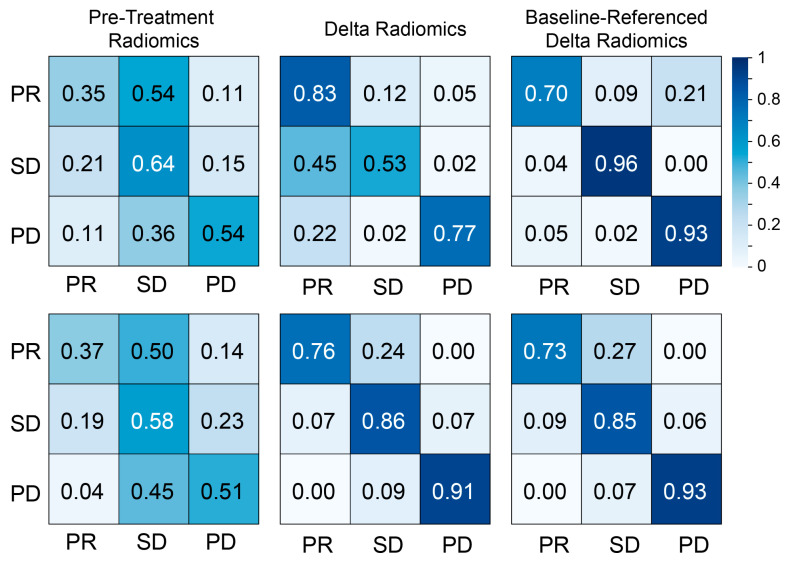
Confusion matrices for pretreatment radiomics, Delta radiomics, and functional radiomics response assessment models computed for patients with liver metastasis from (**top**) colorectal cancer and (**bottom**) breast cancer.

**Figure 5 tomography-11-00020-f005:**
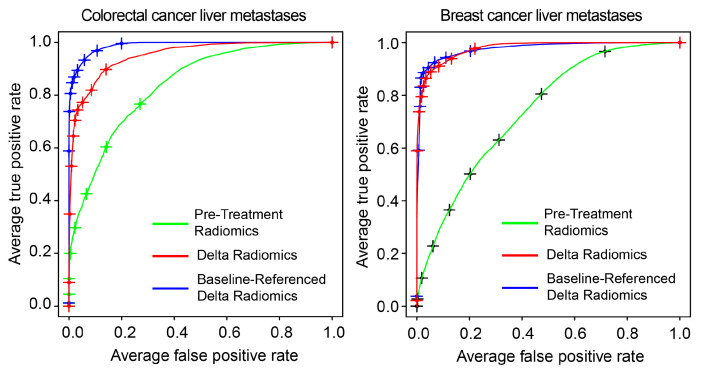
Receiver operating characteristic (ROC) curves for predicting chemotherapy response in liver metastases from colorectal cancer (**left**) and breast cancer (**right**) using pretreatment radiomics, Delta radiomics, and baseline-referenced Delta radiomics.

**Figure 6 tomography-11-00020-f006:**
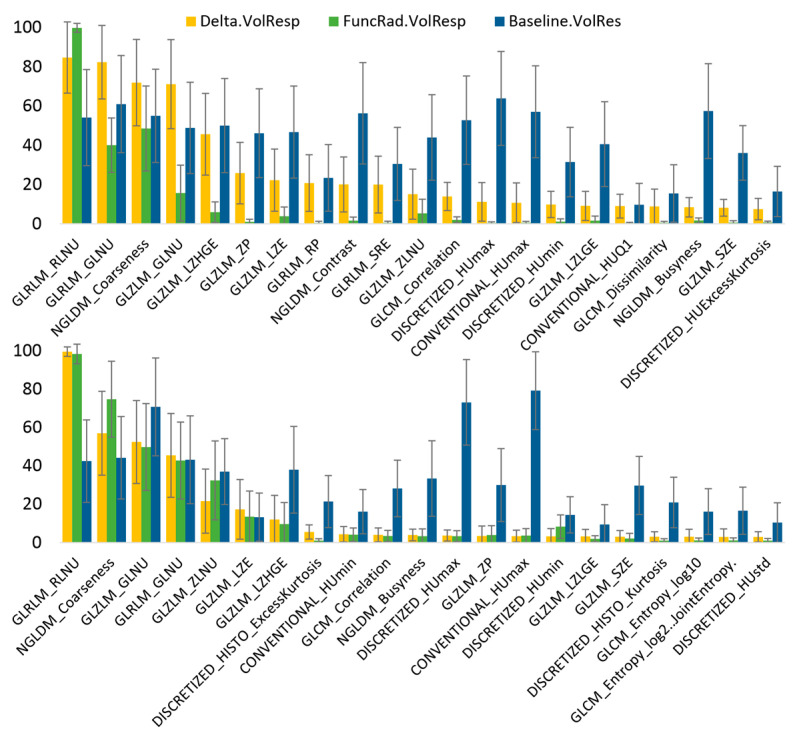
Feature significance for pretreatment radiomics, Delta radiomics, and baseline-referenced Delta radiomics response assessment models computed for patients with liver metastasis from (**top**) colorectal cancer and (**bottom**) breast cancer.

**Figure 7 tomography-11-00020-f007:**
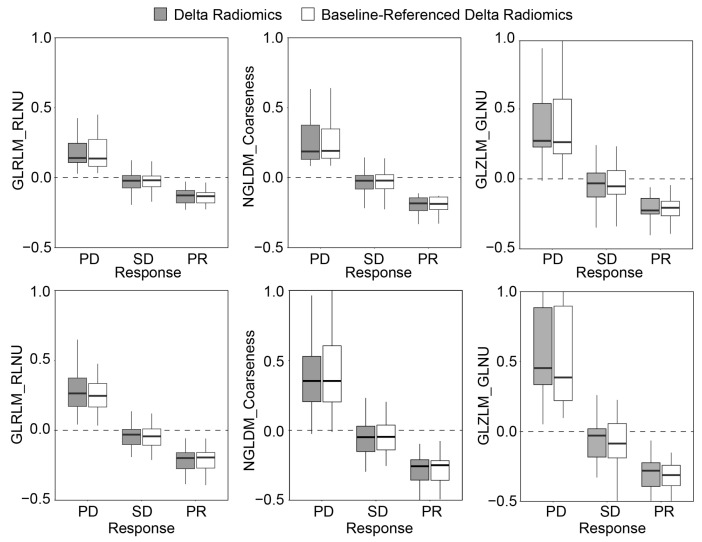
Change in the value of the three most significant features (Figure 5) for Delta radiomics and baseline-referenced Delta radiomics response assessment models computed for patients with liver metastasis from (**top**) colorectal cancer and (**bottom**) breast cancer.

**Figure 8 tomography-11-00020-f008:**
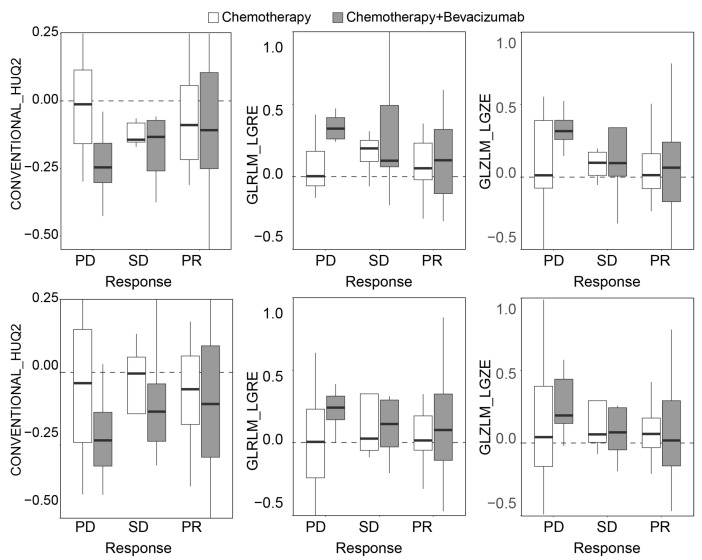
Changes in the value of radiomic features from (**top**) Delta radiomics and (**bottom**) baseline-referenced Delta radiomics models in patients with liver metastasis from colorectal cancer treated with chemotherapy alone or chemotherapy in combination with bevacizumab.

**Table 1 tomography-11-00020-t001:** Patient characteristics, scanner models, and lesion size statistics for each dataset.

	Data Harmonization Dataset				Response Assessment Dataset	
Category	Breast Cancer	Colorectal Cancer	Hepatic Cysts	Liver Parenchyma	Breast Cancer	Colorectal Cancer
Female/male, count	83/–	42/42	132/96	395	57/–	19/18
Female, age	57 ± 13	60 ± 13	62 ± 14	59 ± 13	57 ± 13	60 ± 13
Male, age	–	60 ± 14	67 ± 10	64 ± 13	–	59 ± 13
Subjects, count	83	84	228	395	57	37
Sensation 64	10	13	45	90	3	7
Somatom Drive	20	43	35	91	14	24
Somatom Force	48	69	61	116	29	37
Somatom Definition AS	50	50	55	240	43	26
Somatom Definition	17	14	50	96	7	12
Somatom X.cite	10	3	34	74	10	3
Lesions, count	155	192	280	707	106	109
Diameter, mean ± std, cm	2.5 ± 1.3	2.9 ± 1.5	2.5 ± 1.3	2.4 ± 1.2	2.3 ± 1.3	3.1 ± 1.5
Diameter, median (IRQ), cm	2.1 (1.7–2.7)	2.5 (1.9–3.4)	2.1 (1.6–2.8)	2.7 (1.9–3.5)	2 (1.5–2.6)	2.2 (2–3.2)
Volume, mean ± std, mL	14.5 ± 30.4	26 ± 74.2	15.1 ± 34.6	11.9 ± 10.6	10.7 ± 25	31.3 ± 60.3
Volume, median (IRQ), mL	4.2 (2.5–9.5)	7.5 (3.6–16.6)	4.0 (1.8–11.3)	9.2 (4.51–16.3)	3.3 (1.6–7.6)	4.65 (8.7–27)
Follow-up period, median (IRQ), days	–	–	–	–	108.5 (78–131)	75.0 (58–103)

**Table 2 tomography-11-00020-t002:** Random Forest model performance for pretreatment radiomics, Delta radiomics, and functional radiomics response assessment models computed for patients with colorectal cancer and breast cancer liver metastasis.

			Sensitivity	Specificity	PPV	NPV	Balanced Accuracy
**Colorectal Cancer**	**Pretreatment** **Radiomics**	**PD**	0.25 ± 0.24	0.98 ± 0.03	0.63 ± 0.40	0.93 ± 0.02	0.62 ± 0.12
		**SD**	0.90 ± 0.08	0.21 ± 0.14	0.77 ± 0.03	0.45 ± 0.27	0.56 ± 0.07
		**PR**	0.18 ± 0.17	0.93 ± 0.06	0.37 ± 0.33	0.86 ± 0.02	0.56 ± 0.08
	**Delta** **Radiomics**	**PD**	0.71 ± 0.27	0.97 ± 0.03	0.79 ± 0.22	0.97 ± 0.03	0.84 ± 0.13
		**SD**	0.45 ± 0.21	0.92 ± 0.06	0.55 ± 0.22	0.90 ± 0.03	0.68 ± 0.10
		**PR**	0.88 ± 0.07	0.55 ± 0.17	0.86 ± 0.04	0.63 ± 0.16	0.71 ± 0.08
	**Baseline-referenced Delta Radiomics**	**PD**	0.88 ± 0.10	0.97 ± 0.04	0.93 ± 0.09	0.96 ± 0.04	0.93 ± 0.05
		**SD**	0.97 ± 0.05	0.95 ± 0.06	0.96 ± 0.04	0.95 ± 0.07	0.96 ± 0.04
		**PR**	0.76 ± 0.22	0.94 ± 0.04	0.70 ± 0.20	0.96 ± 0.04	0.85 ± 0.11
**Breast Cancer**	**Pretreatment** **Radiomics**	**PD**	0.45 ± 0.16	0.82 ± 0.09	0.52 ± 0.15	0.79 ± 0.05	0.64 ± 0.08
		**SD**	0.67 ± 0.13	0.43 ± 0.13	0.59 ± 0.07	0.52 ± 0.12	0.55 ± 0.08
		**PR**	0.25 ± 0.18	0.90 ± 0.07	0.37 ± 0.27	0.86 ± 0.03	0.58 ± 0.09
	**Delta** **Radiomics**	**PD**	0.87 ± 0.11	0.96 ± 0.04	0.91 ± 0.09	0.95 ± 0.04	0.91 ± 0.06
		**SD**	0.87 ± 0.09	0.82 ± 0.11	0.86 ± 0.07	0.84 ± 0.09	0.84 ± 0.06
		**PR**	0.73 ± 0.22	0.95 ± 0.05	0.76 ± 0.17	0.95 ± 0.04	0.84 ± 0.10
	**Baseline-referenced Delta Radiomics**	**PD**	0.89 ± 0.10	0.97 ± 0.04	0.93 ± 0.08	0.96 ± 0.04	0.93 ± 0.05
		**SD**	0.88 ± 0.08	0.81 ± 0.10	0.85 ± 0.06	0.85 ± 0.08	0.84 ± 0.06
		**PR**	0.66 ± 0.21	0.94 ± 0.05	0.73 ± 0.18	0.94 ± 0.04	0.80 ± 0.10

## Data Availability

Data are contained within the article and Appendix A.

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
