# Peer review of "Delta Radiomics and Tumor Size: A New Predictive Radiomics Model for Chemotherapy Response in Liver Metastases from Breast and Colorectal Cancer"

_tomography, 2025, doi:10.3390/tomography11030020_

Round 1
Reviewer 1 Report
Comments and Suggestions for Authors
1. In the methodology section, there was lack of a detailed explanation of the baseline-referenced Delta-radiomics model, especially clarifying how the radiomic features were selected and what specific criteria were used for the inclusion in the model.
2. I think the methodology section should provide the specific processing steps used for harmonization of the imaging data from different medical centers and address potential biases introduced by varied scanner settings.
3. Given that this was a retrospective analysis, the methodology section should articulate more explicitly population heterogeneity and statistical power.
4. In the Discussion section, I suggest the authors to discuss the implications of the limitations of the study design on the study’s overall results. For instance, consider how the varied treatment regimens, particularly in the breast cancer cohort, may influence the conclusions drawn regarding chemotherapy response.
5. The discussion should add more on the practical applicability of the baseline-referenced Delta-radiomics model in clinical settings. What specific changes in treatment planning could arise from the use of this model? Please contextualize the findings within current clinical practices.
6. Please include additional figures that illustrate the flowchart processes of Delta radiomics used in the study to enhance comprehension.
Author Response
Comment 1: In the methodology section, there was lack of a detailed explanation of the baseline-referenced Delta-radiomics model, especially clarifying how the radiomic features were selected and what specific criteria were used for the inclusion in the model.
Answer 1: We thank the reviewer for this important comment. We rewrote the entire paragraph on page 7 to improve clarity and address all related questions.
We also added several comments to provide further context about the model. First, we explained the role of the Pretreatment model on page 7: The Pretreatment model, using only baseline radiomics features, serves as a valuable reference point for evaluating the performance of more complex response assessment models like the Delta radiomics. While the Pretreatment model provides a baseline understanding of the predictive power of pre-treatment features alone, it's important to note that it does not predict treatment response on its own. Instead, it helps to establish the incremental value of incorporating post-treatment features, as done in the Delta radiomics model, for improved response assessment.
In addition to Random Forest, the new version of our manuscript includes three other models. We expanded the model descriptions and addressed the question of how the radiomic features were selected and what specific criteria were used for their inclusion in the models (page 8): …All available radiomic features extracted from the post-treatment scans, in conjunction with their corresponding baseline references, were used to generate the baseline-referenced Delta-radiomics features. The rationale behind this approach is to normalize the post-treatment radiomic features with respect to each patient's baseline status for that feature, allowing for a more personalized assessment of change.
The Random Forest, LogitBoost, svmRadial, and pcaNNet models were used to develop classification models for predicting response to chemotherapy and assessing feature importance. Each model incorporates all available transformed features (i.e., the pre-treatment features for the Pretreatment model, the Delta radiomics features for the Delta radiomics model, and the baseline-referenced Delta-radiomics features for the baseline-referenced Delta-radiomics model). These models employ diverse strategies for feature computation and combination. These models employ diverse strategies for feature computation and combination. Random Forest constructs an ensemble of decision trees, each trained on a random subset of features and data and aggregates their predictions (effectively weighting feature importance based on their contribution to the ensemble). LogitBoost, a boosting algorithm, iteratively builds an ensemble of weak learners (typically decision stumps), weighting them based on their performance and focusing on misclassified instances. svmRadial maps all features to a high-dimensional space using a radial basis function kernel and identifies an optimal hyperplane to separate classes. pcaNNet uses principal component analysis to reduce the dimensionality of the feature space (while still incorporating information from all features) before training a neural network, thereby addressing potential issues with high dimensionality and multicollinearity. Feature importance was assessed using the Loess R-squared coefficient; a higher value indicates greater feature importance in the classification model.
Comment 2: I think the methodology section should provide the specific processing steps used for harmonization of the imaging data from different medical centers and address potential biases introduced by varied scanner settings.
Answer 2: We thank the reviewer for this important comment. We added an additional paragraph with corresponding equations to provide a more detailed explanation of the harmonization step. We also cited the original paper, which outlines all specific processing steps (JCO…). Please see pages 6 and 7 for more details.
To standardize the radiomics features across different CT scanners (Table 1), a linear mixed-effects model (LME) for radiomics data harmonization was applied [16]. LME analysis was performed to assess the impact of three fixed-effect variables: cancer/tissue type (T), scanner (S), and tumor volume (V). The volume was the primary variable of interest. Cancer/tissue type and scanner were treated as fixed factors, influencing both the intercept and slope of the linear model. The LME model can be summarized as follows:
F_{T,\ \ S}=f^R+r_T+r_S+∆fR+∆rT+∆rS∙V (1)
Equation E.1 models the linear relationship between a radiomics feature and tumor volume. This model includes fixed effects, which define the reference group, F^R=f^R+∆fR∙V, where f^R is the intercept and ∆fR is the slope. The model also incorporates random effects for scanner and tissue type, including random intercepts, r_S and r_T, and random slopes, ∆rS and ∆rT.
After estimating all random effects, the radiomics features were harmonized by removing the scanner-related random effects.:
F_T^H=F_{T,S}-r_S-∆rS∙V (2)
where F_T^H represents the harmonized radiomics feature, adjusted to remove scanner effects and referenced to the designated group. This harmonization process removes scanner-related variability, preserving the association between radiomics features and tumor size specific to each cancer type. While this approach facilitates more reliable comparisons and analyses of radiomics data, it provides limited control over the broader range of imaging and other technical parameters.
Comment 3: Given that this was a retrospective analysis, the methodology section should articulate more explicitly population heterogeneity and statistical power.
Answer 3: We agree with the reviewer's comment and have added the following sentence to the manuscript (pages 14–15): While the sample size provides some statistical power for detecting key trends, the variability in patient characteristics and treatment regimens (as detailed in Table 1) could influence the observed associations between radiomic features and treatment response, potentially limiting the generalizability of the findings.
Comment 4: In the Discussion section, I suggest the authors to discuss the implications of the limitations of the study design on the study’s overall results. For instance, consider how the varied treatment regimens, particularly in the breast cancer cohort, may influence the conclusions drawn regarding chemotherapy response.
Answer 4: We thank the reviewer for this important comment. We have added the following comment into the manuscript (page 14): Furthermore, the varied treatment regimens, particularly within the breast cancer cohort, may influence our conclusions regarding chemotherapy response. For example, if one regimen dominates in the dataset, the model's performance might reflect its specific efficacy rather than a generalizable response to chemotherapy. On the other hand, a highly diverse range of treatments could obscure true associations between radiomic changes and response due to the confounding effects of the treatments themselves.
Comment 5: The discussion should add more on the practical applicability of the baseline-referenced Delta-radiomics model in clinical settings. What specific changes in treatment planning could arise from the use of this model? Please contextualize the findings within current clinical practices.
Answer 5: We thank the reviewer for this suggestion. We have added the following comment to the manuscript (page 13): Despite these limitations, this model offers two potential clinical advantages. First, it could enable earlier assessment of treatment response compared to traditional methods like RECIST, potentially leading to more timely treatment adjustments. Second, it could be particularly valuable in resource-limited settings where consistent baseline imaging is not always feasible. For example, in cases where the model predicts a poor response early on, clinicians might consider switching to an alternative therapy or exploring clinical trials. In resource-constrained environments, the ability to assess response without baseline scans could expand access to personalized medicine. It's important to note, however, that the model's performance and clinical utility need to be validated in larger, prospective studies before it can be widely adopted in clinical practice.
Comment 6: Please include additional figures that illustrate the flowchart processes of Delta radiomics used in the study to enhance comprehension.
Answer 6: We agree with the reviewer and have added a schematic figure illustrating the study flowchart. Please see Figure 1.
Reviewer 2 Report
Comments and Suggestions for Authors
This study presents a new model, termed baseline-referenced Delta-radiomics, which incorporates the relationship between radiomic features and tumour size into Delta-radiomics to forecast chemotherapy response in liver metastases from 53 cases of breast cancer (BC) and colorectal cancer (CRC). A retrospective study examined contrast-enhanced computed tomography (CT) scans of 83 breast cancer (BC) patients and 84 colorectal cancer (CRC) patients. A total of 57 BC patients with 106 liver lesions and 37 CRC patients with 109 lesions underwent post-treatment imaging following systemic chemotherapy. Radiomic features were derived from a maximum of three lesions per patient after manual segmentation. The tumour response was evaluated by measuring the longest diameter and categorised according to RECIST 1.1 criteria as progressive disease (PD), partial response (PR), or stable disease (SD). Classification models were created to forecast chemotherapy response utilising solely pre-treatment data, Delta radiomics, and baseline-referenced Delta radiomics.
Generally, the topic is interesting, and the paper is well-written. However, it misses a related work section which is essential. Furthermore, some comments need to be addressed to improve the quality of the paper.
The abstract: Please mention which classification models were used.
Introduction:
Could you please write the contributions and novelty in points?
I advise the authors to add a related work section that discusses previous work in the area.
Please add the structure of the remaining sections of the paper.
Methods
Would you kindly provide more information about the preprocessing, segmentation, feature extraction, and classification techniques applied at each stage of the suggested model?
Please provide the mathematical equations for the techniques employed in feature extraction along with a comprehensive description. Additionally, kindly specify the dimensions of each feature set.
Why did the author not employ augmentation methods?
Why did the author use only random forests for classification? Please justify the choice. I advise the authors to compare more than one classification model.
Experimental Results
Please add the results of more classifiers and perform a comparative analysis
Author Response
Comment 1: Could you please write the contributions and novelty in points?
Answer 1: We agree with the reviewer and have change the introduction (page 3):
Our contribution can be summarized as the following:
• Introduction of a Novel Baseline-Referenced Delta-Radiomics Model: The core novelty lies in proposing a new type of Delta-radiomics model that eliminates the need for individual patient baseline radiomics data. This is achieved through a baseline-referencing approach.
• Utilization of Large Pre-treatment Datasets to Establish Baseline Relationships: The model leverages large datasets of pre-treatment scans to empirically derive the functional relationships between radiomic features and tumor size. This da-ta-driven approach allows for a robust and generalizable baseline reference.
• Integration of Radiomic Feature-Tumor Size relationship into Delta-Radiomics: The study incorporates the functional association between radiomic features and tumor size directly into the Delta-radiomics framework.
• Enabling Delta-Radiomics with Post-Treatment Scans Alone: A significant practical contribution is the model's ability to calculate Delta-radiomics values and assess therapy response using only post-treatment scans. This overcomes a key limitation of traditional Delta-radiomics, which requires both pre- and post-treatment imaging.
• Validation Against Conventional Radiomics Models: The study provides empirical validation by comparing the performance of the novel baseline-referenced Delta-radiomics model against established methods: pre-treatment radiomics and conventional Delta-radiomics. This comparative analysis demonstrates the potential advantages of the proposed approach in a clinically relevant setting (liver metastases and systemic therapy response prediction).
Comment 2: I advise the authors to add a related work section that discusses previous work in the area.
Answer 2: We thank the reviewer for this suggestion. We have added a separate paragraph focused on previous work in the area (page 2):
Liver metastases from breast and colorectal cancer present a significant clinical challenge, and early assessment of response to systemic therapy is crucial for optimal patient management. Delta-radiomics, which analyzes changes in radiomic features between pre- and post-treatment images, offers a promising approach for monitoring and predicting therapy response. By quantifying the changes in tumor characteristics captured through radiomic feature analysis of paired pre- and post-treatment medical images [12-15], this technique has the potential to identify imaging biomarkers that enable earlier and more accurate prediction of treatment outcomes. This, in turn, could lead to more personalized and effective treatment strategies for patients with metastatic breast and colorectal cancer. Several recent studies have explored the application of machine learning and CT delta-radiomics to predict tumor response in colorectal and breast cancer liver metastases treated with chemotherapy or targeted therapy (see Table S3 for a summary of key studies) [16, 17]. For example, Ye et al. [18] demonstrated the potential of delta-radiomics, analyzing changes in CT features before and after chemotherapy, to improve the prediction of progression-free survival in patients with colorectal liver metastases, particularly when combined with clinical characteristics. Their model, based on delta-radiomics and clinical data, outperformed models based on single time-point radiomics. Similarly, Lu et al. [19] showed that a deep learning network can effectively capture tumor morphological changes beyond size, improving early prediction of treatment response to anti-VEGF therapy in metastatic colorectal cancer and outperforming traditional size-based assessments. Integrating this deep learning approach with conventional size measurements further enhanced predictive accuracy, suggesting its value for personalized early treatment decisions in metastatic colorectal cancer. These findings, while promising, highlight the need for further research to validate the potential of delta-radiomics for early response assessment and personalized treatment strategies, particularly in breast cancer liver metastases.
Comment 3: Please add the structure of the remaining sections of the paper.
Answer 3: We thank the reviewer for this suggestion. We have added a concluding paragraph describing the manuscript's structure (page 3): This paper first establishes the Materials and Methods, then describes the response assessment models, including the novel baseline-referenced Delta-Radiomics model. It details the development of the predictive model, and the evaluation methods used to assess its performance. Finally, the paper concludes by discussing the results, clinical implications, future directions, and limitations of this project.
Comment 4: Would you kindly provide more information about the preprocessing, segmentation, feature extraction, and classification techniques applied at each stage of the suggested model?
Answer 4: We thank the reviewer for valuable comments and suggestions regarding the Methods. We have added multiple changes. Please see updated Methods and Materials (pages 4-9)
Comment 5: Please provide the mathematical equations for the techniques employed in feature extraction along with a comprehensive description. Additionally, kindly specify the dimensions of each feature set.
Answer 5: We thank the reviewer for this important comment. In our work, we adhere to the Image Biomarker Standardization Initiative (IBSI) and utilize LIFEx software, which is compatible with this standard. We have also specified the dimensions of each feature set and included a list of all radiomic features used in our study in Supplemental Table S4. We have added the necessary comments and references to the text. The following references have been included:
• Zwanenburg, A., M. Vallières, M.A. Abdalah, H. Aerts, V. Andrearczyk, A. Apte, S. Ashrafinia, S. Bakas, R.J. Beukinga, R. Boellaard, M. Bogowicz, L. Boldrini, I. Buvat, G.J.R. Cook, C. Davatzikos, A. Depeursinge, M.C. Desseroit, N. Dinapoli, C.V. Dinh, S. Echegaray, I. El Naqa, A.Y. Fedorov, R. Gatta, R.J. Gillies, V. Goh, M. Götz, M. Guckenberger, S.M. Ha, M. Hatt, F. Isensee, P. Lambin, S. Leger, R.T.H. Leijenaar, J. Lenkowicz, F. Lippert, A. Losnegård, K.H. Maier-Hein, O. Morin, H. Müller, S. Napel, C. Nioche, F. Orlhac, S. Pati, E.A.G. Pfaehler, A. Rahmim, A.U.K. Rao, J. Scherer, M.M. Siddique, N.M. Sijtsema, J. Socarras Fernandez, E. Spezi, R. Steenbakkers, S. Tanadini-Lang, D. Thorwarth, E.G.C. Troost, T. Upadhaya, V. Valentini, L.V. van Dijk, J. van Griethuysen, F.H.P. van Velden, P. Whybra, C. Richter, and S. Löck, The Image Biomarker Standardization Initiative: Standardized Quantitative Radiomics for High-Throughput Image-based Phenotyping. Radiology, 2020. 295(2): p. 328-338.
• Zwanenburg, A., Leger, S., Vallières, M., and Löck, S. (2016). Image biomarker standardisation initiative - feature definitions. In eprint arXiv:1612.07003 [cs.CV]
• C. Nioche, F. Orlhac, S. Boughdad, S. Reuzé, J. Goya-Outi, C. Robert, et al. Cancer research 2018 Vol. 78 Issue 16 Pages 4786-4789
Comment 6: Why did the author not employ augmentation methods?
Answer 6: We thank the reviewer for this interesting question. We agree that augmentation techniques, such as applying various imaging filters (e.g., Gaussian, Laws, Wavelets), could potentially enhance radiomics models. However, these techniques are not yet standardized and are not included in the IBSI guidelines. Moreover, several studies have shown that radiomic features computed from filtered images often exhibit low importance due to strong correlations with the original radiomic features. While we acknowledge the potential benefits of augmentation, we chose not to include it in this initial study for these reasons.
A separate comment was added to address this limitation (page 16): In this study, we did not employ imaging filters such as Gaussian, Laws, or Wavelets. Future work could explore the impact of data augmentation on the relationship between radiomic features and tumor size, as well as investigate optimal combinations of augmentation techniques and feature selection methods. This would improve the robustness and generalizability of Delta radiomics models across diverse datasets and imaging protocols.
Comment 7: Why did the author use only random forests for classification? Please justify the choice. I advise the authors to compare more than one classification model. Experimental Results: Please add the results of more classifiers and perform a comparative analysis.
Answer 7: We thank the reviewer for this important question. We acknowledge that our initial choice of classification model limits the generalizability of our findings, and a more comprehensive analysis of other models is necessary. We have now added results for three additional models: LogitBoost (a boosting classification algorithm), svmRadial (Support Vector Machines with a Radial Basis Function kernel), and pcaNNet (a neural network with principal component analysis preprocessing). These additional models yielded similar outcomes, with results varying within a range of approximately 10%. We have included these results in the revised manuscript (pages 8–9) and added three new supplemental tables (Tables S5–S7) showing the performance of each model in response assessment for patients with colorectal and breast cancer liver metastases. Furthermore, we have expanded our discussion (page 15).
Reviewer 3 Report
Comments and Suggestions for Authors
The article is a retrospective study, and I believe it has the following limitations:
1. The number of breast cancer and colorectal cancer patients selected in the article is relatively small, and there may be external influences among patients such as treatment methods, race, region, and physical fitness, which will limit the generalizability of the study.
2. The article uses a linear mixed model to standardize radiomic features, but differences in imaging protocols and equipment can still have an impact on the experimental data, potentially affecting the accuracy of the data and the results.
3. The article only uses CT technology to detect the tumor conditions of patients. However, CT technology cannot fully display all characteristics of tumors, and some radiological features may not be well represented by CT, potentially leading to an incomplete assessment of the tumors. Additionally, because the study is retrospective, it is not possible to systematically control variables to obtain experimental data, which may lead to selection bias and affect the results.
4. The assessment in this experiment relies on the RECIST 1.1 criteria, but the article has already mentioned that this standard has certain limitations, which may introduce some bias into the experimental results.
Author Response
Comment 1: The number of breast cancer and colorectal cancer patients selected in the article is relatively small, and there may be external influences among patients such as treatment methods, race, region, and physical fitness, which will limit the generalizability of the study.
Answer 1: We thank the reviewer for this important comment. We agree with the reviewer and have addressed this limitation with an additional comment (page 16): Additionally, the relatively small sample size, coupled with potential external influences such as variations in treatment methods, patient demographics (including race, region, and physical fitness), limits the generalizability of our findings and warrants further investigation in larger, more diverse cohorts.
Comment 2: The article uses a linear mixed model to standardize radiomic features, but differences in imaging protocols and equipment can still have an impact on the experimental data, potentially affecting the accuracy of the data and the results.
Answer 2: We thank the reviewer for this important point. We agree that variations in imaging protocols and equipment can impact radiomics data and potentially affect the accuracy of our results. Due to the retrospective nature of this study, we did not have access to complete details regarding imaging protocols, which likely contributes to the variability in radiomics features observed in Figure 2, particularly in the association with tumor size. While our linear mixed model provides a degree of standardization, we acknowledge that it offers limited control over these technical variations. A more comprehensive standardization, incorporating factors such as acquisition parameters, and reconstruction algorithms, would require a significantly larger and more meticulously curated dataset to ensure sufficient statistical power for each factor. Future prospective studies should prioritize the collection of detailed imaging metadata to enable more robust harmonization and analysis. We added an additional sentence (Page 7) to highlight this limitation: It is important to note that while this approach facilitates more reliable comparisons and analyses of radiomics data, it provides limited control over the broader range of imaging and other technical parameters.
Comment 3: The article only uses CT technology to detect the tumor conditions of patients. However, CT technology cannot fully display all characteristics of tumors, and some radiological features may not be well represented by CT, potentially leading to an incomplete assessment of the tumors. Additionally, because the study is retrospective, it is not possible to systematically control variables to obtain experimental data, which may lead to selection bias and affect the results.
Answer 3: We agree with the reviewer on this point and thank for the comment. While computed tomography (CT) is one of the most common imaging techniques in clinical practice, especially for response assessment in cancer patients, it does have limitations in characterizing liver lesions. This manuscript presents a novel radiomics model for response assessment in patients with liver metastases. The advantage of the radiomics approach is its potential to provide a more comprehensive and personalized analysis, as demonstrated by our findings regarding differential responses to therapy in CRC patients. However, the inherent challenges of retrospective studies often limit the ability to control for all potentially confounding variables. These limitations include, but are not limited to, variations in patient demographics, treatment protocols, and image acquisition parameters across different institutions. Future prospective studies with standardized protocols and larger, more diverse patient cohorts are needed to validate these findings and further explore the potential of this radiomics model. We modified the limitations paragraph to address this comment (Page 15): Our study had several limitations. First, it was a retrospective study with a limited sample size and heterogeneous treatment regimens, especially within the BC cohort. This retrospective design, while reflecting real-world clinical practice, introduces the possibility of selection bias and limits our ability to fully control for confounding variables. Second, we only considered CT scans of liver metastases from BC and CRC. While CT is a readily available and commonly used imaging modality for assessing liver metastases, it has inherent limitations in fully characterizing all tumor features. Certain radiological characteristics may not be optimally represented on CT, potentially leading to an incomplete assessment of the tumor response to therapy. In future, the model analysis should be further expanded to include other imaging modalities, such as MRI and PET, which may provide complementary information.
Comment 4: The assessment in this experiment relies on the RECIST 1.1 criteria, but the article has already mentioned that this standard has certain limitations, which may introduce some bias into the experimental results.
Answer 4: We thank the reviewer for this comment. While RECIST 1.1 is currently the most common criterion used to assess response to therapy, incorporating other clinical factors would certainly provide a more comprehensive assessment. However, this is beyond the scope of the present study. We agree that future radiomics studies should include demographic, longitudinal imaging, and treatment details.
Round 2
Reviewer 2 Report
Comments and Suggestions for Authors
I would like to thank the reviewers for addressing most of my comments. However, the authors have not addressed my question properly: Would you kindly provide more information about the preprocessing, segmentation, feature extraction, and classification techniques applied at each stage of the suggested model? Please state briefly in both the rebuttal letter before going the details of the steps and methods used in the proposed framework
Author Response
We thank the reviewer for raising this important point. We agree that a clear and detailed description of the data preprocessing steps is crucial for reproducibility and ensuring the validity of our results. We have reorganized this section to ensure that each step is presented in the correct order and that all details are provided. Briefly, our radiomics framework involves five main steps: (1) CT Preprocessing, where we resampled all CT scans to a uniform resolution; (2) Tumor Segmentation and Response Assessment, where we manually delineated the lesions and determined response to therapy according to RECIST 1.1 criteria; (3) CT Radiomic Feature Extraction, where we collect all radiomic features; (4) CT Radiomic Feature Harmonization, where we removed scanner-associated bias; and (5) Model Building and Statistical Analysis, where we trained machine learning models. The attached file provides a detailed description of each stage.

Reviewer 3 Report
Comments and Suggestions for Authors
The authors have addressed most of my concerns, I recommend acceptance.
Author Response
Thank you for you time and help!